# High-Dose Electron Radiation and Unexpected Room-Temperature Self-Healing of Epitaxial SiC Schottky Barrier Diodes

**DOI:** 10.3390/nano9020194

**Published:** 2019-02-02

**Authors:** Guixia Yang, Yuanlong Pang, Yuqing Yang, Jianyong Liu, Shuming Peng, Gang Chen, Ming Jiang, Xiaotao Zu, Xuan Fang, Hongbin Zhao, Liang Qiao, Haiyan Xiao

**Affiliations:** 1Institute of Nuclear Physics and Chemistry, China Academy of Engineering Physics, P.O. Box 919-220, Mianyang 621900, China; biansechong@163.com (G.Y.); pyl641122@163.com (Y.P.); yangsapphire@aliyun.com (Y.Y.); m13568266352@163.com (J.L.); pengshuming@caep.cn (S.P.); 2State Key Laboratory of Wide-Bandgap Semiconductor Power Electronics, Nanjing 210000, China; steelchg@163.com; 3School of Physical, University of Electronic Science and Technology of China, Chengdu 610054, China; mjianglw@gmail.com (M.J.); xtzu@uestc.edu.cn (X.Z.); 4School of Science, Changchun University of Science and Technology, 7089 Wei-Xing Road, Changchun 130022, China; fangxuan110@126.com; 5State Key Laboratory of Advanced Materials for Smart Sensing, General Research Institute for Nonferrous Metals, Beijing 100088, China; zhaohongbin@grinm.com

**Keywords:** electron irradiation, room temperature self-healing, noise, electron-induced current, I–V curve

## Abstract

Silicon carbide (SiC) has been widely used for electronic radiation detectors and atomic battery sensors. However, the physical properties of SiC exposure to high-dose irradiation as well as its related electrical responses are not yet well understood. Meanwhile, the current research in this field are generally focused on electrical properties and defects formation, which are not suitable to explain the intrinsic response of irradiation effect since defect itself is not easy to characterize, and it is complex to determine whether it comes from the raw material or exists only upon irradiation. Therefore, a more straightforward quantification of irradiation effect is needed to establish the direct correlation between irradiation-induced current and the radiation fluence. This work reports the on-line electrical properties of 4H-SiC Schottky barrier diodes (SBDs) under high-dose electron irradiation and employs in situ noise diagnostic analysis to demonstrate the correlation of irradiation-induced defects and microscopic electronic properties. It is found that the electron beam has a strong radiation destructive effect on 4H-SiC SBDs. The on-line electron-induced current and noise information reveal a self-healing like procedure, in which the internal defects of the devices are likely to be annealed at room temperature and devices’ performance is restored to some extent.

## 1. Introduction

Silicon carbide (SiC) is an ideal material for high-frequency, high-power, and high-temperature devices due to its strong hardness, excellent thermal/mechanical stability, high-thermal conductivity, and controlled electrical properties [1,2]. With the advance of state-of-the-art growth and epitaxy techniques in the past decade, a variety of high-quality SiC-based structures and devices have been fabricated [1], making them very competitive toward third-generation wide-gap semiconductors. Thus, SiC-based structures have recently regained significant attention in advanced device applications. Furthermore, owing to its large intrinsic fundamental band gap (up to 3.26 eV [3]), SiC is not suitable to visible and infrared light stimulation or irradiation, and their radiation resistance is stronger than that of the conventional semiconductors (such as Si-based materials); therefore, SiC-based diodes are very promising candidates to replace silicon devices for atomic battery light sensors, nuclear cell photosensors [4,5], or particle detectors [6,7].

Required by long-term exposure applications, the radiation resistance and durability of SiC diodes (as radiation detectors and atomic battery light sensors) are very critical and have been subjected to extensive research. For particle detector applications, the detectivity and radiation resistance of SiC diodes were studied under gamma ray [7,8], proton [7,8], neutron [9], electron [8,10], and X-ray [6] irradiations. The results indicated that the electrical properties of radiated SiC diodes and their radiation resistance depended on types, energies, flux, fluence, and the absorbed dose of irradiated particles. It was found that when the total dose of 32-MeV proton was 8.5 × 10^12^ cm^−2^ and the total absorbed dose of 1.25-MeV γ-ray was 1000 kGy (Air), the electrical properties of 6H-SiC diodes were basically unchanged [7]. Within the range of 1–10 Gy absorbed dose, the radiation damage to 4H-SiC diodes caused by 22-MeV electron and 6-MeV X-ray photon beams hardly affected the radiation-induced current [6]. When the neutron fluence exceeded 5.7 × 10^16^ cm^−2^, the charge collection efficiency of the 4H-SiC diode started to drop [9]. For atomic battery light sensor applications, electrons were an important radiation source [4]; thus, many studies had focused on the ability of 4H-SiC diodes’ resistance to electron irradiation. For example, Eiting et al. studied the radiation resistance of 4H-SiC p-i-n junction betavoltaic irradiated with 8.5 GBq ^33^P source with an average beta decay energy of 77 keV, and found that the devices did not experience degradation under the irradiation of ^33^P source for four half-lives of ^33^P source, thus demonstrating a good electron radiation resistance [4]. Chandrashekhar et al. experimentally studied the devices’ radiation resistance performance under the 1 mCi ^63^Ni β-radiation source for ten days, and the results suggested such irradiation conditions did not cause the degradation of 4H-SiC diodes [5].

In practical applications for both radiation detectors and atomic battery light sensors in many extreme conditions, such as nuclear, aerospace, military, and astrophysics, the total expected irradiation dose is normally enormous. However, the most common research of experimental or theoretical study on SiC electron radiation resistance is still relatively low; for example, the largest total 8.2-MeV electron fluence ever reported is 9.48 × 10^14^ cm^−2^ [8,10]. Large fluence of electron irradiation is not only common in the radiation detection experiments, but also critical in the integrated design of the next-generation atomic battery, and thus deserves further study.

Moreover, the general principle of particle detector and atomic battery light sensor is based on induced-current generation (by radiation of electrons, neutrons, photons, and heavy ions) and detection (of the induced-voltage signals in diodes). The previous research were mostly focused on the correlation between the internal defects of SiC materials (or devices) and their electrical parameters under electron irradiation, such as relationship between defect levels and I–V as well as C–V characteristic curves [5,8]. However, this apparent relationship is not suitable to explain the intrinsic response of irradiation effect, since defects are not easy to characterize and it is complex to distinguish whether the defects come from the raw material or exists only upon irradiation. Therefore, a more straightforward quantification of irradiation effect is greatly needed to establish the direct correlation between irradiation-induced current and the radiation fluence.

For this purpose, low-frequency noise information, especially 1/f noise, can be used as a characterization means for the sensitive detection of internal defects in electronic materials [11] and devices [12,13]. This method in principle can also be utilized to characterize radiation damage to electronic devices [12]. Under the beam radiation, the internal defects and the structural damage will lead to the increase of low-frequency noise power spectral density *S_V_* [14,15]. For example, Babcock et al. [16] studied the radiation resistance of Ultra High Vacuum/Chemical Vapor Deposition SiGe heterojunction bipolar transistor (HBT) irradiated by ^60^Co γ-ray and found that the increase of the total absorbed dose resulted in the performance degradation of the SiGe HBT. The current gain *β* was decreased with the increase of absorbed dose, and the internal defects were increased, while the noise information *S_V_* increased in the low-frequency range, as compared with that before irradiation. Therefore, the 1/f noise can be used as a tool to characterize the defects in materials and devices, and correlate the defects with devices performance.

Here, we reported the study of electron-induced current and radiation resistance of SiC Schottky barrier diodes (SBDs) under high-fluence electron irradiation as well as an in situ noise diagnostics for defect–electrical property analysis. Through on-line electron-induced current, I–V curve, noise information, and SBDs’ radiation resistance to the environment of electron irradiation had been analyzed, and a self-driven healing process was observed at room temperature, which led to some extent of electrical performance recovery.

## 2. Materials and Methods

### 2.1. 4H-SiC SBDs Samples and Irradiation Experimental Conditions

The epitaxial 4H-SiC SBDs used in this experiment were provided by State Key Laboratory of Wide-Band Gap Semiconductor Power Electronics located at Nanjing, China. Its structure is shown in Figure 1a, where the photosensitive area of SBDs is 3 mm × 3 mm and the voltage-withstand range is −100 to 100 V. The 4H-SiC SBDs were 4H-SiC epilayers grown by chemical vapor deposition on SiC substrates of 360 mm thickness, with nitrogen doped with a net doping density of 1 × 10^18^ cm^−3^ and micropipe density of 1 micropipe cm^−2^. The buffer was n-type, 1 µm thick, with a net free carrier concentration of 1 × 10^18^ cm^−3^. The epilayer was n-type, 12 µm thick, with a net free carrier concentration of 3 × 10^15^ cm^−3^. The ohmic contact was obtained by deposition of a 1000-Å-thick layer of Ni and a 3-µm-thick layer of Au. The Schottky contact was obtained by radio frequency magneton sputtering of a 1000-Å-thick layer of Ni at room temperature. In order to reduce the influence of environment on the device, SiO_2_ of 1000 Å thickness and Si_3_N_4_ of 1000 Å thickness were grown on Ni by sputtering method.

Electron beam irradiation experiments were performed on the electron accelerator of Sichuan Forever Holding Co., Ltd, located at Mianyang, China. The electron energy was 1.8 MeV, the electron flux was 9.62 × 10^12^ cm^−2^ s^−1^, and the total electron fluence was 9.05 × 10^17^ cm^−2^. In order to avoid overheating of SiC SBDs during electron beam irradiation, the bottom of the irradiated metal platform was continuously cooled with water, and meanwhile, the temperature of the SiC SBDs was monitored by a thermocouple and controlled at a constant temperature of 25 °C. The home-made real-time on-line current test system was used to monitor and record the changes of the electron-induced current of SiC SBDs during irradiation and 30 min after electron irradiation. Then the devices were kept at room temperature (25 °C) for 72 h without heating. The devices performance tended to decrease with the increasing electron fluence, but it was not certain whether the devices were disabled or not. In the case of reverse breakdown, the devices would be disabled completely, and the reverse breakdown state can be regarded as the worst-case state of devices performance. The devices were reverse biased to breakdown at 200 V voltage by Keithley 6517B high impedance/electrometer located at Mianyang, China toward the end of the experiment, and the damage of the device after irradiation was evaluated when the devices performance at reverse breakdown state was considered as the worst-case state. The I–V curves and noise information of SiC SBDs before and after electron beam irradiation, room-temperature self-healing, and reverse breakdown were measured by Keithley 2635 sourcemeter and the self-developed noise parameter test system located at Mianyang, China.

### 2.2. The Electron-Induced Current Test

When subjected to radiations of electron beam, γ-ray, and neutron, current can be induced in SBDs due to the ionization effect within Schottky barrier junctions, which is shown in Figure 1b. The principle of the electron-induced current can be understood by taking the electrons as an example—1.8 MeV electrons penetrate the SBDs completely. The n-type 4H-SiC material of the Schottky barrier junction was irradiated by incident electrons with an energy larger than the material’s fundamental electronic band gap; thus electron–hole pairs would be generated in n-type 4H-SiC material. The Fermi lever in the Ni Schottky contact was less than the Fermi lever in the n-type 4H-SiC material; therefore, the average energy of electrons in the n-type 4H-SiC material was greater than the average energy of those in the Ni Schottky contact. The difference in the average electron energy can be expected to transfer electrons from the n-type 4H-SiC material to the Ni Schottky contact until the average electron energies were equal. While the holes can only stay in n-type 4H-SiC material, the electrons were pulled toward the Ni Schottky contact. As a result, the width of the depletion region became narrower, and the contact potential difference decreased. The incident electron energy was converted into electrical energy. Once the external circuit was short-circuited, current could flow through the Schottky barrier junction. In this way, the separation of the electron–hole pairs can be achieved, and the induced current *I_P_*, whose direction was from the Ni Schottky contact to the n-type 4H-SiC material through the Schottky barrier junction, can be produced. The current equation of Schottky barrier junction under electron beam irradiation is
(1)$$I=I_{0}(e^{eU/kT}-1)-I_{P}$$where *I* and *I*_0_ are the current flowing through the Schottky barrier junction and the reverse saturation current, respectively; e, *U*, and *T* stand for the electron charge, the applied voltage, and temperature, respectively; and *k* is Boltzmann constant (*k* = 1.38 × 10^−23^ J K^−1^).

When the applied voltage *U* is 0 V, the Formula (1) becomes
*I* = −*I_P_*.(2)

In this study, the open-circuit induced current of SiC SBDs has been tested by a real-time on-line current test system, as shown in Figure 2a. This test system consisted of the fixtures, low-loss cables, 10-channel scanning card, data interface, Keithley 6517B high impedance/electrometer, and self-compiled control software (installed in the control computer). The fixtures were used to fix SiC SBDs and transmit current signals to low-loss cables. The 10-channel scanning card was installed at the rear panel of Keithley 6517B high impedance/electrometer and they were used to capture multiple electron-induced currents. The self-compiled control software was used to control and record current data detected by Keithley 6517B high impedance/electrometer in real time. The system can achieve a 10-channel real-time signal acquisition with the current and voltage accuracies of each signal acquisition reaching way up to 1 fA and 1 nV, respectively.

### 2.3. Noise Information Test

In this study, the changes of noise information before and after irradiation of SiC SBDs have been tested by using the noise parameter test system. The noise parameter test system consists of low-noise bias, adapter, Stanford Research Systems SR560 voltage amplifier, acquisition card, and XD3020 [17] noise analysis software with the system bandwidth being 0–10^6^ Hz, the background noise being <4 nV/√Hz (@1 kHz), and the signal magnification capability being 10^0^–10^5^, as shown in Figure 2b.

The noise of the diode was typically composed of two or three components of shot noise, 1/f noise, and generation-recombination noise (G-R noise). The noise power spectral density (*S*) can be written as:
(3)$$S\left(f\right)=A+\frac{B}{f^{\gamma }}$$where *A* is the shot noise amplitude, *B*, the 1/f noise amplitude, and the exponents of *γ* is usually taken to be unity.

In this work, based on the noise spectrum information of SiC SBDs, the electron irradiation effects of SiC SBDs were studied by using the parameter information in Equation (3). The damage degree of the device had been determined according to the change of noise power spectral density before and after irradiation and room-temperature self-healing.

### 2.4. I–V Curve Test

The current–voltage (I–V) curve of a semiconductor device contained the electrical information of dark current as well as break-over voltage. The real-time on-line electron-induced current and I–V characteristics can characterize the device to some extent. The I–V curve with voltage range of −100 to 2.5 V before and after irradiation of SiC SBDs had been measured with Keithley 2635 sourcemeter in this work. The damage degree of the device had been determined according to the change of I–V curve before and after irradiation and room-temperature self-healing.

## 3. Results and Discussion

### 3.1. High-Dose Electron Irradiation

Figure 3 shows the real-time curves of the open-circuit electron-induced current in SiC SBDs. It can be seen that when the energy of electrons was 1.8 MeV, and the electron flux was 9.62 × 10^12^ cm^−2^ s^−1^, a current of 1.27 × 10^−5^ A was induced due to the ionization effect. However, as the electron fluence further increased, the electron-induced current of the device decreased continuously. When the electron fluence reached 1 × 10^15^ cm^−2^, the current decreased to 1.24 × 10^−5^ A, accounting for only a decrease of 2.36%, which was consistent with the results reported by Nava et al. with the total electron fluence being 9.48 × 10^14^ cm^−2^ [8,10]. SiC SBDs can be considered as sufficient to resist the irradiation of electrons with a fluence under 1 × 10^15^ cm^−2^. However, with a further increase of electron fluence, the electron-induced current of SiC SBDs began to decrease sharply, particularly when the electron fluence was ≤3 × 10^17^ cm^−2^. When the electron fluence was 1 × 10^16^ cm^−2^, the current decreased to 1.03 × 10^−5^ A, showing a decrease of 18.90%. When the electron fluence was 1 × 10^17^ cm^−2^, the current decreased to 5.00 × 10^−6^ A, accounting for a decrease of 60.63%. At the end of irradiation, the total electron fluence reached 9.05 × 10^17^ cm^−2^ and the current became 1.82 × 10^−6^ A, showing a decrease of 85.70%. It was noteworthy that when the electron fluence was >3.87 × 10^17^ cm^−2^, the electron-induced current fluctuation of SiC SBDs increased sharply and the current fluctuation could reach up to 86%.

When SiC SBDs were irradiated by high dose of electrons, a large number of defects would be generated inside the silica layer [18], silicon carbid layer [19,20,21,22], and the interfaces [23,24] between the metal and silicon carbide. These defects would produce a removal effect [25,26] on charge carriers at 1.8 MeV electron irradiation, leading to the decrease of current.

Hemmingsson et al. studied the deep level defects in electron-irradiated 4H-SiC epitaxial layers grown by chemical vapor deposition using deep level transient spectroscopy (DLTS). The measurements performed on electron-irradiated p-n junctions in the temperature range 100–750 K revealed both electron and hole traps with thermal ionization energies ranging from 0.35 to 1.65 eV [27], which led to the deterioration of device performance. Iwamoto et al. investigated the formation and evolution of defects in 4H-SiC Schottky barrier diodes and correlated with the SBDs’ performances [28]. The SBDs were irradiated with 1 MeV electrons to a fluence of 1.00 × 10^15^ cm^−2^. Current–voltage, capacitance–voltage, and DLTS measurements were used to study the effect of defects on the SBDs performance. It was found that the DLTS defect levels (EH_1_, EH_3_, and Z_1/2_) were very likely to be partly responsible for the charge collection efficiency reduction after electron irradiation. EH_1_ and EH_3_ were related to carbon interstitials and Z_1/2_ was related to carbon vacancies. DLTS study on n-type 4H-SiC (0001) epilayers also showed that the carrier lifetime in an n-type 4H-SiC epilayer, measured by differential microwave photoconductance decay, had been significantly improved from 0.73 μs (as-grown) to 1.62 μs (after oxidation, 1300 °C) as the Z_1/2_ and EH_6/7_ centers had been reduced from (0.3–2) × 10^13^ cm^−3^ to below the detection limit (1 × 10^11^ cm^−3^) by thermal oxidation of epilayers at 1150–1300 °C [29]. Based on the above results, we suggested that the current fluctuation might due to the increasing defects in the devices, such as interstitials and vacancies. These defects led to a slower and less stable carrier motion, which resulted in current fluctuation.

Figure 4 shows the I–V curves (including reverse breakdown region) of SiC SBDs before and after irradiation. From Figure 4b, it can be seen that the reserve current of SiC SBDs before irradiation increases with the increase of reserve bias, showing typical diode characteristics. For the 2-h irradiation sample, the forward current of the device dropped sharply from 2.39 × 10^−5^ A (before irradiation) to 2.06 × 10^−8^ A at a voltage of 2.5 V, while the absolute value of the reverse dark current further increased (Figure 2b). When the voltage became −100 V, the current was −8.76 × 10^−7^ A, showing a serious degradation of the device characteristics. At the reverse breakdown of the device, the I–V curve of the device had been tested. It was found that the forward current was further reduced compared to the case of 2 h after irradiation. When the voltage became 2.5 V, the current dropped to 1.42 × 10^−9^ A, while the voltage became −100 V, the current was −1.3 × 10^−9^ A, which was similar to the I–V characteristics of the resistance. The electron capture levels had been proved to be induced by the electron irradiation, and it would drastically influence the resistance in the bulk crystal and the Schottky barrier of SiC SBDs [30]. Under 2 MeV electron irradiation, the Schottky barrier was found to decrease and the resistance of the bulk crystal was found to increase with the electron fluence, leading to the decrease of the forward current. The Schottky barrier of SiC SBDs decreased from 1.25 eV (pre-irradiation) to 1.17 eV (post-irradiation, 1.00 × 10^17^ cm^−2^). The decrease of the barrier height was thus responsible for the increase of the reverse current. On the other hand, the increase of electron-induced defects in passivation layers was expected to result in the increase of the reverse leakage [31,32]. In this work, the I–V curves (2 h after electron irradiation) exhibited similar behavior. The SiC SBDs could be considered as the resistance since the barrier height disappeared and the bulk crystal resistance increased further after the SiC SBDs were reverse biased to breakdown. The I–V curves (reverse breakdown) of SiC SBDs were similar to the I–V characteristics of the resistance.

Besides the electrical properties (electron-induced current and I–V curves), the noise information of SiC SBDs also reflects the internal defect states before and after irradiation and the reverse breakdown, as shown in Figure 5a. It can be seen that in the frequency range of 10^2^ Hz to 10^5^ Hz, the noise power spectral density, *S_V_*, exhibited a clear dependence on the electron irridation, with *S_V_* (reverse breakdown) > *S_V_* (2 h after irradiation) > *S_V_* (pre-irradiation). Ziel et al. [33] proposed that under small fluence irradiation, induced current was mainly contributed by two components: (1) the current $I_{0}e^{qU/kT}$ due to the injection of electrons from the semiconductor into the metal and (2) –*I*_0_ due to the injection of electrons from the metal into the semiconductor. Both currents contained carriers that pass independently and randomly through the junction barrier, thereby showing as pure intermediate frequency shot noise. In this work, the shot noise of current SiC SBDs was in the frequency range of 10^2^ to 10^5^ Hz.

The current noise power spectral density (*S_I_*) of SBDs’ shot noise is [34]
(4)$$S_{I}=2e\left(I+2I_{0}\right)$$

The SBDs resistance can be obtained by Equation (1)
(5)$$R=\frac{dU}{dI}\approx \frac{kT}{e\left(I+I_{0}\right)}$$

The voltage noise power spectral density *S_V_* is
(6)$$A=S_{V}=S_{I}\times R^{2}=\frac{2{\left(kT\right)}^{2}\left(I+2I_{0}\right)}{e{\left(I+I_{0}\right)}^{2}}$$

The current *I* is
(7)$$I=\frac{2ne}{t}$$where *R* stands for the diode resistance, *n* and *t* are the carrier concentration and time for drifting through the barrier, respectively.

When $I\gg I_{0}$, Equation (6) can be simplified as
(8)$$S_{V}\approx \frac{2{\left(kT\right)}^{2}}{eI}=\frac{t{\left(kT\right)}^{2}}{e^{2}n}$$

As the current decreased, *n* decreased, which made *S_V_* to increase according to Equation (8), thus leading to *S_V_* (2 h after irradiation) > *S_V_* (pre-irradiation). When SiC SBDs first experienced the electron beam irradiation and then subjected to reverse breakdown, a large number of defects would be produced inside SiC SBDs, which made the noise power spectral density of SiC SBDs to further increase compared with SBDs that only irradiated by electrons, therefore showing *S_V_* (reverse breakdown) > *S_V_* (2 h after irradiation).

### 3.2. Room-Temperature Self-Healing

As shown in Figure 5a, the noise spectra of SiC SBDs before and after irradiation as well as in reverse breakdown state are 1/f noise in the frequency range of 10^0^ to 10^2^ Hz. 1/f noise had an extrinsic origin that arose from defects [35]. The defects of electronic origin lying in the gap acted as electron traps leading to both mobility and carrier density fluctuations. In the frequency range of 10^0^ to 10^2^ Hz, *S_V_* reached the minimum before irradiation, while reached the maximum at 2 h after electron irradiation, and *S_V_* even was greater than that of the reverse breakdown. However, if the SiC SBDs were left alone for 72 h after electron irradiation at room temperature, their *S_V_* could become lower than that of *S_V_* at reverse breakdown state, showing a significant decrease. The observed *S_V_* (2 h after irradiation) > *S_V_* (72 h after irradiation) was partially due to the fact that part of the defects were rapidly annealed at room temperature; thus, the effect of carrier removal was weakened, leading to the increase of carrier concentration. According to Hooge’s equation [36], the relationship between *S_V_* and carrier concentration *n* are given by Equation (9).
(9)$$S_{V}\left(f\right)=\frac{B}{f^{\gamma }}\propto \frac{\alpha_{H}}{n}$$where $\alpha_{H}$ stands for the Hooge parameter. The *S_V_* corresponding to 1/f noise was inversely proportional to the carrier concentration inside the device. As carrier concentration increased, *S_V_* decreased.

The noise power spectra at 2 h and 72 h after irradiation (shown in Figure 5a) cannot clearly reflect the annealing effect of the internal defects in SiC SBDs at room temperature. Therefore, at 72 h after irradiation exposure, we have carried out the tracking measurement on the low-frequency noise of SiC SBDs irradiated by electron beam at room temperature, which is shown in Figure 5b. It can be clearly seen that in the frequency range of 10^0^~10^2^ Hz, the noise power spectral density of SiC SBDs decreased with time, and the observable frequency range of 1/f noise gradually decreased. At 2 h after irradiation, 1/f noise can be observed in the frequency range of 1–100 Hz; at 4 h after irradiation exposure, 1/f noise can be observed in the frequency range of 1–30 Hz; and at 72 h after irradiation exposure, 1/f noise frequency further decreased into 1–6 Hz, while 1/f noise will be annihilated by shot noise above these frequency ranges.

The temporal variation of 1/f noise amplitude of the SiC SBDs irradiated by electrons can be calculated by Equation (9), and the results are shown in Figure 5c, where the 1/f noise amplitude *B* decreases exponentially with time *t*, and this relationship is further well fitted by Equation (10).
(10)$$B=7.81\times 10^{-13}e^{-t/0.81}+2.50\times 10^{-15}$$where the unit of *B* is V^2^·Hz^−1^, the unit of *t* is s.

Then the declining rate is
(11)$$\frac{dB}{dt}=-9.64\times 10^{-13}e^{-t/0.81}$$

At 4.5 h after irradiation, *B* declined sharply, while the declining rate decreased with the increase of time, and the declining rate followed Equation (11). At 72 h after irradiation, the declining rate tended to be constant. The value of *B* decreased from 6.75 × 10^−14^ V^2^·Hz^−1^ (2 h after irradiation) to 2.05 × 10^−15^ V^2^·Hz^−1^ (72 h after irradiation), accounting for 96.96% decrease.

Figure 5b,c shows that the internal defects of the SiC SBDs continuously decrease at room temperature, weakening the removal effects of the defects on carriers. Back in 1966, Fischerrr et al. [37] had systematically investigated the temperature annealing of traps produced by 6–88 MeV electron irradiation in n-type Ge. In their experiment, the electron irradiation temperature was 85 K, and it was found that some traps disappeared when the temperature increased to near 200 K, thereby, leading to an increase of the carrier concentration. In 2009, Messina et al. [38] found that a significant portion of $E_{\gamma }^{\prime }$ centers, which induced in amorphous silica at room temperature by γ-irradiation up to 79 kGy, spontaneously decayed after the end of irradiation. In 1984, Yamaguchi et al. [39] found that effective room-temperature self-healing of radiation-induced defects in both p-type and n-type InP after electron irradiation leading to the recovery of InP solar cell properties. Besides Ge, amorphous silica, and InP solar cell, room-temperature self-healing of radiation-induced defects were also shown in Si-based devices [25,40,41,42,43], as reported by Pease et al. Pease et al. [25] found the normalized change in reciprocal resistivity in the drain to source ($\bar{∆1/R_{DS}}$) of Si-based power Metal-Oxide-Semiconductor Field-Effect Transistors degraded between 2 and 10% (most less than 5%) over a period of 24 h after room-temperature proton and neutron irradiation. These phenomena suggested that the internal defects generated inside the semiconductor material under irradiation can be annealed and annihilated below room temperature. However, as far as the room-temperature annealing of SiC materials is concerned, currently there is still not enough experimental evidences to demonstrate that some defects of SiC materials can be annealed after electron irradiation at room temperature. In this work, we used low-frequency noise to show that electron irradiation can produce the defects in SiC devices and these induced defects can be further annealed at room temperature. Assuming only electron irradiation and *α_H_* remained constant, regardless of the amount of electron irradiation, part of internal defects in SiC material at 72 h after irradiation can be calculated to decrease by 3.04% as compared with the same internal defects at 2 h after irradiation. Since 1/f noise cannot be directly used to differentiate the types of defects, it is impossible to know what kinds of defects are annealed at room temperature. However, this property can still be used to characterize the declining trend of internal defects with time after electron irradiation via the noise information, and this method is also applicable for the defects characterization in other semiconductor materials and devices.

Furthermore, noise characterization were used to demonstrate that the SiC SBDs after electron irradiation have undergone an unexpected self-annealing or relaxation process at room temperature, thereby, reducing the induced defects while increasing the carrier concentration. Such room-temperature annealing was consistent with the on-line electron-induced current curve after irradiation and vice versa. As shown in Figure 6, the data collected by the on-line current–voltage test system shows that the electron-induced current of the device does not decrease to the background level within 19 min after the exposure to electron irradiation, but instead presents a linear increasing profile with the current rising from 10^−10^ to 10^−7^ A. This phenomenon suggested that within 19 min after the irradiation, the device was subjected to a room-temperature annealing or relaxation, curing some of the internal defects, and recovering the electrical performance to a certain level. This phenomenon was also consistent with the I–V characteristics of the SiC SBDs, demonstrating that although the performance of the SiC SBDs was greatly lowered after the exposure to 9.05 × 10^17^ cm^−2^ electron irradiation, this type of semiconductor device can still recover its electrical properties to some extent under this unexpected self-annealing mechanism.

## 4. Conclusions

The electrical performance of SiC SBDs irradiated by high-dose, high-energy (1.8 MeV) electron beam had been investigated in this work. It was found that electron beam had a strong radiation destructive effect on 4H-SiC SBDs. Their electrical performance was greatly reduced when exposed to high-energy electron beam with a fluence as high as 9.05 × 10^17^ cm^−2^, the electron-induced current reduced by 85.70%, and the device characteristics degraded seriously. The on-line electron-induced current and noise information revealed a self-healing like procedure, in which the internal defects of the devices were likely to be annealed at room temperature and devices performance was restored to some extent. Although the mechanism of this self-healing is under investigation, the high-dose irradiation and noise diagnostics study reported here can provide useful information for designing next-generation atomic battery and detectors for extreme environment applications.

## 5. Patents

Guixia Yang, Yuanlong Pang, Fansong Zeng, et al. Low-noise bias device, the Chinese patent of invention, patent number: ZL 201510729381.1.

## Figures and Tables

**Figure 1 nanomaterials-09-00194-f001:**
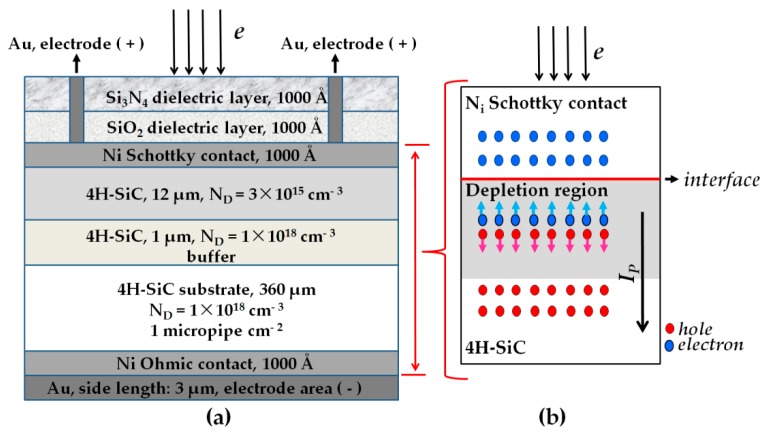
(**a**) The schematic diagram of Silicon carbide (SiC) Schottky barrier diodes (SBDs) and (**b**) the principle of electron-induced current generation under irradiation.

**Figure 2 nanomaterials-09-00194-f002:**
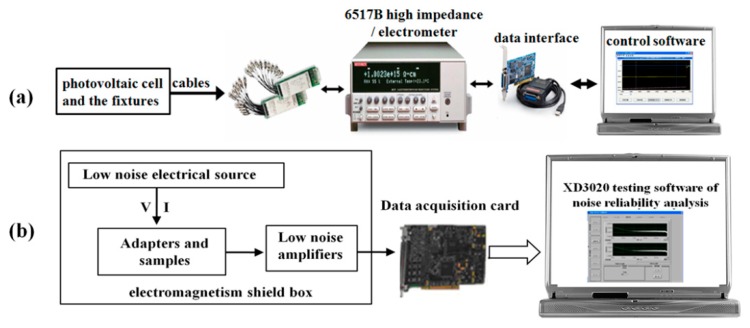
The illustration for (**a**) real-time on-line current test system and (**b**) test system for noise parameters.

**Figure 3 nanomaterials-09-00194-f003:**
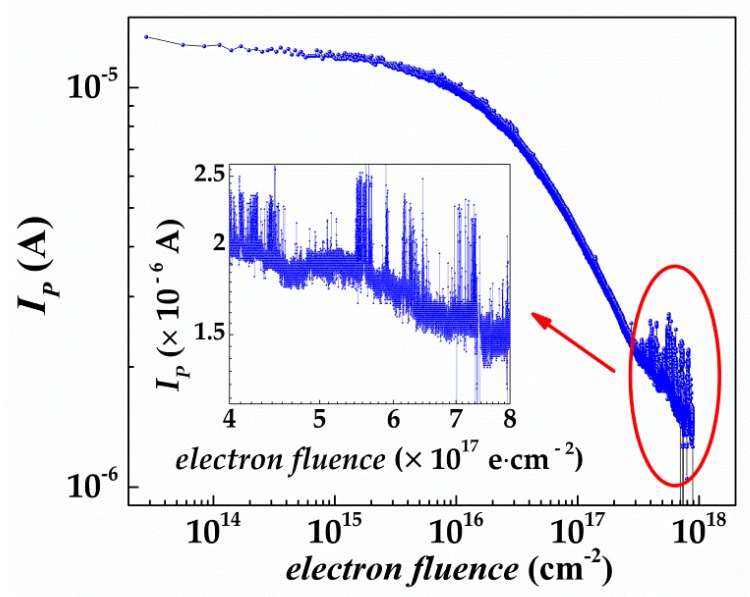
The real-time changing profiles of the induced current with electron fluence.

**Figure 4 nanomaterials-09-00194-f004:**
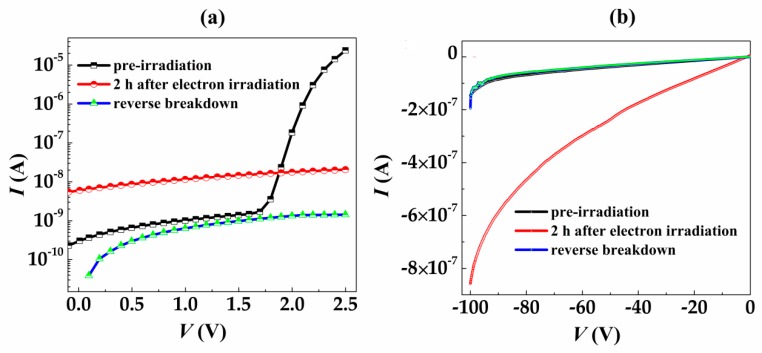
The I–V curves for SiC SBDs before and after electron beam irradiation. (**a**) The foward bias region and (**b**) the reverse bias region.

**Figure 5 nanomaterials-09-00194-f005:**
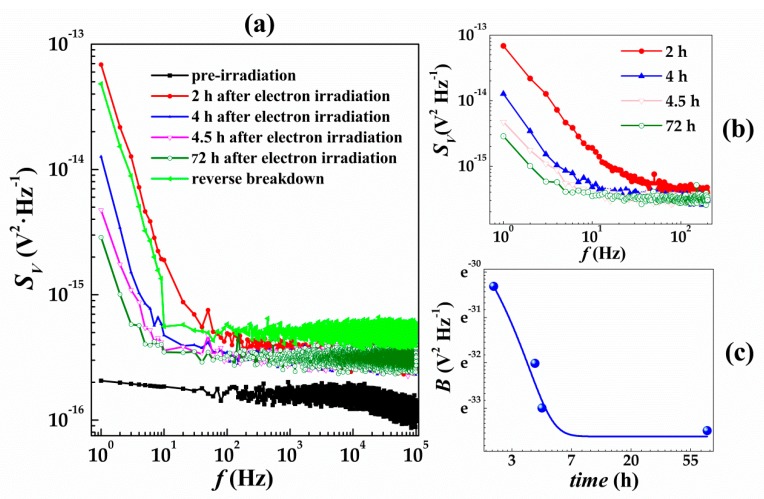
The noise power spectral densities of SiC SBDs. (**a**) Before and after electron beam irradiation and in a reverse breakdown state; (**b**) the low-frequency noise curve within 72 h after electron irradiation; and (**c**) the changing profiles of the low-frequency noise amplitude within 72 h after electron irradiation.

**Figure 6 nanomaterials-09-00194-f006:**
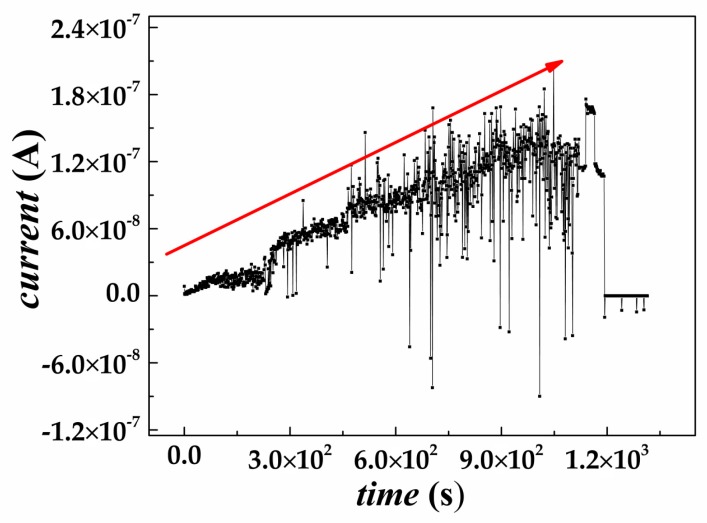
The real-time electron-induced current curve of SiC SBDs after the end of exposure to electron beam irradiation.

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
