# Peer review of "High-Dose Electron Radiation and Unexpected Room-Temperature Self-Healing of Epitaxial SiC Schottky Barrier Diodes"

_nanomaterials, 2019, doi:10.3390/nano9020194_

Round 1

Reviewer 1 Report

The authors presented an interesting electrical characterization of SiC Schottky diodes.
However, the soundness of the paper must be improved introducing the identification of the defects responsible of the presented noise phenomenon.

In particular, the authors claimed that "the current fluctuation is mainly due to the increasing vacancies and defects in the device". Si or C vacancies or what else? This sentence has to be clarified. The paper needs to to explain which are the defects responsible of the described behavior. DLTS and/or structural characterization and/or adequate refences are needed to discriminate among point and extended defect; Si or C vacancies etc.

Concerning the description of Fig 4, Can the author exclude any trapping phenomenon in the SiO2 passivation layer?

Please avoid the contract form "doesn't"

Author Response

Dear reviewer:

We thank for your careful examination of the manuscript, and for your helpful comments that helped improve its quality. We have addressed all major and minor issues that were raised. Please find below our replies in a point-by-point fashion:

Comment 1.English language and style: Moderate English changes required.

Reply to Comment 1: Thank you for your valuable comment. English language and style has be modified.

Comment 2.Does the introduction provide sufficient background and include all relevant references? Must be improved.

Reply to Comment 2: Thank you for your valuable comment. The introducing the identification of the defects responsible of the presented noise phenomenon was added to the revised manuscript.

“For this purpose, low frequency noise information, especially 1/f noise, can be used as a characterization means for the sensitive detection of internal defects in electronic materials [11] and devices [12,13]. This method in principle can also be utilized to characterize radiation damage to electronic devices [12]. Under beam radiation, the internal defects and the structural damage will lead to the increase of low-frequency noise power spectral density SV [14,15]. For example, Babcock [16] et al. studied the radiation resistance of UHV/CVD SiGe HBT irradiated by 60Co γ-ray and found that the increase of the total absorbed dose results in the performance degradation of the SiGe HBT The current gain β was decreased with the increase of absorbed dose, internal defects were increased, while the noise information SV increased in the low frequency range, as compared with that before irradiation. Therefore, the 1/f noise can be used as tool to characterize the defects in materials and devices, and correlate defects with devices performance.” The above discussion was added to the revised manuscript, Page 2, line 49. (see Introduction).

Comment 3.Are the conclusions supported by the results? Must be improved.

Reply to Comment 3: Thank you for your valuable comment. The conclusions has be modified (see Conclusions).

Comment 4.The authors presented an interesting electrical characterization of SiC Schottky diodes. However, the soundness of the paper must be improved introducing the identification of the defects responsible of the presented noise phenomenon.

Reply to Comment 4: We havd added the discussion on defects responsible of the presented noise phenomenon in the revised manuscript, Page 2, line 49. (see Introduction).

Comment 5.In particular, the authors claimed that "the current fluctuation is mainly due to the increasing vacancies and defects in the device". Si or C vacancies or what else? This sentence has to be clarified. The paper needs to to explain which are the defects responsible of the described behavior. DLTS and/or structural characterization and/or adequate refences are needed to discriminate among point and extended defect; Si or C vacancies etc.

Reply to Comment 5: Thank you for your valuable comment. We suggest that the current fluctuation observed at high fluences (figure 3) might due to the increasing defects in the device, such as interstitials and vacancies. The related statement of the manuscript is not clear. We have revised the statement, and provide more evidence in the revised manuscript.

"Hemmingsson et al. studied the deep level defects in electron-irradiated 4H SiC epitaxial layers grown by chemical vapor deposition using deep level transient spectroscopy (DLTS). The measurements performed on electron-irradiated p-n junctions in the temperature range 100–750 K revealed both electron and hole traps with thermal ionization energies ranging from 0.35 to 1.65 eV[27], which leads to the deterioration of device performance. Iwamoto et al. investigated the formation and evolution of defects in 4H-SiC Schottky barrier diodes and correlated with the SBDs’ performances [28]. The SBDs were irradiated with 1 MeV electrons to a fluence of 1.00×1015 cm-2. Current-voltage, capacitance-voltage, and DLTS measurements were used to study the effect of defects on the SBDs performance. It was found that the DLTS defect levels (EH1, EH3 and Z1/2) were very likely to be partly responsible for the charge collection efficiency reduction after electron irradiation. EH1 and EH3 are related to carbon interstitials and Z1/2 is related to carbon vacancies. DLTS study on n-type 4H-SiC (0001) epilayers also showed that the carrier lifetime in an n-type 4H-SiC epilayer, measured by differential microwave photoconductance decay, had been significantly improved from 0.73μs (as-grown) to 1.62μs (after oxidation: 1300 ℃) as the Z1/2 and EH6/7 centers has been reduced from (0.3-2)×1013 cm-3 to below the detection limit (1×1011 cm-3) by thermal oxidation of epilayers at 1150 –1300 ℃[29]. Based on the above results, we suggest that the current fluctuation might due to the increasing defects in the devices, such as interstitials and vacancies. These defects lead to slower and less stable carrier motion, which results in current fluctuation." The above discussion was added to the revised manuscript, Page 6, line 27.

Comment 6.Concerning the description of Fig 4, Can the author exclude any trapping phenomenon in the SiO2 passivation layer?

Reply to Comment 6: The trapping phenomenon in the SiO2 passivation layer should be included for the description of Fig 4. We suggest that the defects in the SiO2 passivation layer can increase reverse leakage of SiC SBDs. As requested by the reviewer, the discussion is provided.

"On the other hand, the increase of electron induced defects in passivation layers is expected to result in increase of the reverse leakage [31,32]." The above discussion was added to the revised manuscript, Page 7, line 33.

Comment 7.Please avoid the contract form "doesn't".

Reply to Comment 7: The "doesn't" have be changed. (see Page 1, line 24; Page 2, line 4; Page 2, line 19; Page 2, line 43)

Reviewer 2 Report

General Comments:

 The current fluctuations observed at high fluences (figure 3) is attributed to the generation of vacancies in SiC. On what basis is this statement made? More evidence should be provided.

I-V characteristics before and after irradiation are presented. How was the reverse breakdown treatment done? Comments about the observed I-V characteristics should be provided. Why is the I-V behavior so different between the pre and post irradiation?

Why were the SBDs subjected to breakdown after irradiation?

Specific Comments:

Page 2, line 8: the referencing is not correct (don't list the first authors of different citaitons in this manner)

The previous literature concerning SiC robustness to various types of radiation should be preferably tabulated the energy as well as fluence should always be provided

Page 3: Figure 2a & b should be Figure 1a & b

Page 3, line 19: " the device is annealed at room temperature for 72 hours" What does this mean exactly?

Was the SBD biased during the irradiation?

The following sentence is not clear: " In order to compare the degree of damage, the SBDs were subjected to reverse breakdown at 200V voltage after annealing at room temperature for 72 h, which made it completely lose the function of the device." please clarify.

Page 9: Define Δ1/RDS

The formatting of some references needs to be checked.

Author Response

Dear reviewer:

We thank for your careful examination of the manuscript, and for your helpful comments that helped improve its quality. We have addressed all major and minor issues that were raised. Please find below our replies in a point-by-point fashion:

Comment 1.Does the introduction provide sufficient background and include all relevant references? Must be improved.

Reply to Comment 1: Thank you for your valuable comment. We have added some background discussion regarding identification of the defects responsible of the presented noise phenomenon in the revised manuscript. Changes are listed here:

“For this purpose, low frequency noise information, especially 1/f noise, can be used as a characterization means for the sensitive detection of internal defects in electronic materials [11] and devices [12,13]. This method in principle can also be utilized to characterize radiation damage to electronic devices [12]. Under beam radiation, the internal defects and the structural damage will lead to the increase of low-frequency noise power spectral density SV [14,15]. For example, Babcock [16] et al. studied the radiation resistance of UHV/CVD SiGe HBT irradiated by 60Co γ-ray and found that the increase of the total absorbed dose results in the performance degradation of the SiGe HBT The current gain β was decreased with the increase of absorbed dose, internal defects were increased, while the noise information SV increased in the low frequency range, as compared with that before irradiation. Therefore, the 1/f noise can be used as tool to characterize the defects in materials and devices, and correlate defects with devices performance.” The above discussion was added to the revised manuscript, Page 2, line 49. (see Introduction).

Comment 2.Are the conclusions supported by the results? Must be improved.

Reply to Comment 2: Thank you for your valuable comment. The conclusions has be modified (see Conclusions).

Comment 3.The current fluctuations observed at high fluences (figure 3) is attributed to the generation of vacancies in SiC. On what basis is this statement made? More evidence should be provided.

Reply to Comment 3: We suggest that the current fluctuation observed at high fluences (figure 3) might due to the increasing defects in the device, such as interstitials and vacancies. The related statement of the manuscript is not clear. We have revised the statement, and provide more evidence in the revised manuscript.

"Hemmingsson et al. studied the deep level defects in electron-irradiated 4H SiC epitaxial layers grown by chemical vapor deposition using deep level transient spectroscopy (DLTS). The measurements performed on electron-irradiated p-n junctions in the temperature range 100–750 K revealed both electron and hole traps with thermal ionization energies ranging from 0.35 to 1.65 eV[27], which leads to the deterioration of device performance. Iwamoto et al. investigated the formation and evolution of defects in 4H-SiC Schottky barrier diodes and correlated with the SBDs’ performances [28]. The SBDs were irradiated with 1 MeV electrons to a fluence of 1.00×1015 cm-2. Current-voltage, capacitance-voltage, and DLTS measurements were used to study the effect of defects on the SBDs performance. It was found that the DLTS defect levels (EH1, EH3 and Z1/2) were very likely to be partly responsible for the charge collection efficiency reduction after electron irradiation. EH1 and EH3 are related to carbon interstitials and Z1/2 is related to carbon vacancies. DLTS study on n-type 4H-SiC (0001) epilayers also showed that the carrier lifetime in an n-type 4H-SiC epilayer, measured by differential microwave photoconductance decay, had been significantly improved from 0.73μs (as-grown) to 1.62μs (after oxidation: 1300 ℃) as the Z1/2 and EH6/7 centers has been reduced from (0.3-2)×1013 cm-3 to below the detection limit (1×1011 cm-3) by thermal oxidation of epilayers at 1150 –1300 ℃[29]. Based on the above results, we suggest that the current fluctuation might due to the increasing defects in the devices, such as interstitials and vacancies. These defects lead to slower and less stable carrier motion, which results in current fluctuation." The above discussion was added to the revised manuscript, Page 6, line 27.

Comment 4.I-V characteristics before and after irradiation are presented. How was the reverse breakdown treatment done? Comments about the observed I-V characteristics should be provided. Why is the I-V behavior so different between the pre and post irradiation?

Reply to Comment 4: Thank you for your valuable comment. The reverse breakdown treatment: the voltage -withstand range of the 4H-SiC SBDs is -100 V - 100 V, so the devices were reverse biased to breakdown at 200V voltage by Keithley 6517B high impedance / electrometer at last of experiment. Comments about the observed I-V characteristics have been provided.

"The electron capture levels has been proved to be induced by the electron irradiation and it would drastically influence the resistance in the bulk crystal and the Schottky barrier of SiC SBDs [30]. Under 2 MeV electron irradiation, the Schottky barrier was found to decrease and the resistance of the bulk crystal was found to increase with electron fluence, leading to the decrease of the forward current. The Schottky barrier of SiC SBDs has been decreased from 1.25 eV (pre-irradiation) to 1.17 eV (post-irradiation, 1.00×1017 cm-2). The decrease of the barrier height is thus responsible for the increase of the reverse current. On the other hand, the increase of electron induced defects in passivation layers is expected to result in increase of the reverse leakage [31,32]. In this work, the I-V curves (2-hour after electron irradiation) exhibits similar behavior. The SiC SBDs could be considered as the resistance since the barrier height disappeared and the bulk crystal resistance increased further after the SiC SBDs were reverse biased to breakdown. The I-V curves (reverse breakdown) of SiC SBDs is similar to I-V characteristics of the resistance." The above discussion was added to the revised manuscript, Page 7, line 26.

Comment 5.Why were the SBDs subjected to breakdown after irradiation?

Reply to Comment 5: The devices performance tended to decrease with the increasing electron fluence, but it was not certain whether the devices were disabled or not. In the case of reverse breakdown, the devices would be disabled completely, and the reverse breakdown state can be regarded as the worst-case state of devices performance. So the devices were biased to breakdown at 200V voltage at last of experiment, and the damage degree of the device after irradiation is evaluated when the devices performance at reverse breakdown state was considered as the worst-case state.

Comment 6.Page 2, line 8: the referencing is not correct (don't list the first authors of different citaitons in this manner)

Reply to Comment 6: Thank you for your valuable comment.

The sentence "S. Metzger, S. Seshadri, F. Nava, and A. Castaldini, et al. studied the SiC diodes’ detectivity and their corresponding resistance under gamma rays, protons, neutrons, electrons and X-rays irradiations." was modified as:

"the detectivity and radiation resistance of SiC diodes’ were studied under gamma rays[7,8], protons[7,8], neutrons[9], electrons[8] [10]and X-rays[6] irradiations. " (see page 2 line 11).

Comment 7.The previous literature concerning SiC robustness to various types of radiation should be preferably tabulated the energy as well as fluence should always be provided.

Reply to Comment 7: Thank you for your valuable comment. The previous literature concerning SiC robustness to various types of radiation have been tabulated the energy as well as fluence.

Comment 8.Page 3: Figure 2a & b should be Figure 1a & b.

Reply to Comment 8: The Figure 2a & b was changed to Figure 1a & b. (see page 4)

Comment 9.Page 3, line 19: " the device is annealed at room temperature for 72 hours" What does this mean exactly?

Reply to Comment 9: "the device is annealed at room temperature for 72 hours" means the device was kept at room temperature (25 ℃) for 72 hours without heating. The sentence was changed to "the devices were kept at room temperature (25 ℃) for 72 hours without heating". (see Page 3 line38)

Comment 10.Was the SBD biased during the irradiation?

Reply to Comment 10: The SBDs were not biased during the irradiation.

Comment 11.The following sentence is not clear: " In order to compare the degree of damage, the SBDs were subjected to reverse breakdown at 200V voltage after annealing at room temperature for 72 h, which made it completely lose the function of the device." please clarify.

Reply to Comment 11: The sentence was modified as: "The devices performance tended to decrease with the increasing electron fluence, but it was not certain whether the devices were disabled or not. In the case of reverse breakdown, the devices would be disabled completely, and the reverse breakdown state can be regarded as the worst-case state of devices performance. The devices were reverse biased to breakdown at 200V voltage by Keithley 6517B high impedance / electrometer at last of experiment, and the damage of the device after irradiation is evaluated when the devices performance at reverse breakdown state was considered as the worst-case state."(see Page 3 line 39)

Comment 12.Page 9: Define Δ1/RDS.

Reply to Comment 12: Δ1/RDS was defined as " the normalized change in reciprocal resistivity in the drain to source of Si-based power MOSFETs". (see Page 10 line 25)

Comment 13.The formatting of some references needs to be checked.

Reply to Comment 13: The references have been modified.

Round 2

Reviewer 2 Report

The authors of the publication below have addressed all of my comments.